

# Comparison of radiomics-based machine-learning classifiers for the pretreatment prediction of pathologic complete response to neoadjuvant therapy in breast cancer

Xue Li, Chunmei Li, Hong Wang, Lei Jiang and Min Chen

Radiology, Beijing Hospital, Beijing, China
Graduate School of Peking Union Medical College, Beijing, China

## ABSTRACT

**Background**. Machine learning classifiers are increasingly used to create predictive models for pathological complete response (pCR) in breast cancer after neoadjuvant therapy (NAT). Few studies have compared the effectiveness of different ML classifiers. This study evaluated radiomics models based on pre- and post-contrast first-phase T1 weighted images (T1WI) in predicting breast cancer pCR after NAT and compared the performance of ML classifiers.

**Methods**. This retrospective study enrolled 281 patients undergoing NAT from the Duke-Breast-Cancer-MRI dataset. Radiomic features were extracted from pre- and post-contrast first-phase T1WI images. The Synthetic Minority Oversampling Technique (SMOTE) was applied, then the dataset was randomly divided into training and validation groups (7:3). The radiomics model was built using selected optimal features. Support vector machine (SVM), random forest (RF), decision tree (DT), k-nearest neighbor (KNN), extreme gradient boosting (XGBoost), and light gradient boosting machine (LightGBM) were classifiers. Receiver operating characteristic curves were used to assess predictive performance.

**Results**. LightGBM performed best in predicting pCR [area under the curve (AUC): 0.823, 95% confidence interval (CI) [0.743–0.902], accuracy 74.0%, sensitivity 85.0%, specificity 67.2%]. During subgroup analysis, RF was most effective in pCR prediction in luminal breast cancers (AUC: 0.914, 95% CI [0.847–0.981], accuracy 87.0%, sensitivity 85.2%, specificity 88.1%). In triple-negative breast cancers, LightGBM performed best (AUC: 0.836, 95% CI [0.708–0.965], accuracy 78.6%, sensitivity 68.2%, specificity 90.0%).

**Conclusion**. The LightGBM-based radiomics model performed best in predicting pCR in patients with breast cancer. RF and LightGBM showed promising results for luminal and triple-negative breast cancers, respectively.

Corresponding authors
Lei Jiang, jiang_belinder@sina.com
Min Chen, cjr.chenmin@vip.163.com

## BACKGROUND

As breast cancer treatment evolves, neoadjuvant therapy (NAT) has become the standard of care for patients with large tumors, certain molecular subtypes, and locally advanced tumors (*Heil et al., 2020*; *Huang et al., 2020*; *Bitencourt et al., 2020*). NAT is effective in reducing tumor size and thus increasing the rate of breast-conserving surgery (*Spring, Bar & Isakoff, 2022*). Additionally, it can decrease axillary lymph node burden, potentially allowing for sentinel lymph node biopsy in place of axillary dissection (*Spring, Bar & Isakoff, 2022*). Pathologic complete response (pCR) is a surrogate endpoint for assessing the efficacy of NAT and is associated with improved clinical outcomes, such as prolonged overall survival (OS) and disease-free survival (DFS) (*Cho et al., 2014*). In contrast, patients who do not achieve pCR generally have a higher risk of relapse (*Waks & Winer, 2019*). In clinical practice, pCR only occurs in a minority of patients, and pCR rates vary according to molecular subtypes; 50–60% of patients achieve pCR in HER2-positive breast cancer (*Gianni et al., 2012*), while only 5–20% achieve pCR in high-grade luminal cancer (*Cortazar et al., 2014*; *Sikov et al., 2015*). Evaluation of pCR mainly relies on the histopathology of postoperative specimens. Thus, the noninvasive and early prediction of pCR, especially before NAT, can help clinicians optimize treatment strategies and avoid unnecessary drug toxicity or disease progression.

Dynamic contrast-enhanced magnetic resonance imaging (DCE-MRI) is the optimal imaging modality for evaluating the efficacy of NAT and has the highest sensitivity compared with mammography and ultrasound (US) (*Mann et al., 2008*; *Fowler, Mankoff & Joe, 2017*). However, a major disadvantage of DCE-MRI in predicting pCR is the low positive predictive value (PPV) (*Kim et al., 2022*). The commonly used DCE-MRI metrics are qualitative morphology-based indicators, which show late-stage changes in tumor microstructure. Quantitative assessment modalities could help improve the diagnostic performance of DCE-MRI in identifying pCR status. Radiomic analysis is an emerging method that has become increasingly popular for predicting pCR to NAT in breast cancer (*Bitencourt et al., 2020*; *Liu et al., 2019*). By evaluating the gray level position and intensity of pixels on medical images and converting them into high-dimensional feature data, radiomic analysis can quantitatively reflect the spatial heterogeneity and microenvironment of tissues. This generates image-driven biomarkers. Radiomics models based on pretreatment DCE-MRI have been shown to be accurate in predicting breast cancer pCR [area under the curve (AUC) 0.707–0.858] (*Yoshida et al., 2022*; *Caballo et al., 2023*; *Cain et al., 2019*; *Peng et al., 2021*). *Cain et al. (2019)* analyzed pre-NAT breast DCE-MRI images in 288 patients with breast cancer, extracted 529 radiomics-based features per patient, and constructed relevant models. The researchers then trained and validated the models using two machine learning (ML) algorithms (logistic regression and support vector machine). Their findings suggested that radiomics models based on pretreatment DCE-MRI can potentially predict pCR in patients with triple-negative or HER2-positive breast cancer.

In recent years, the investigation of ML classifiers to determine the most effective treatment strategies in oncology has gained significant momentum. This study aimed
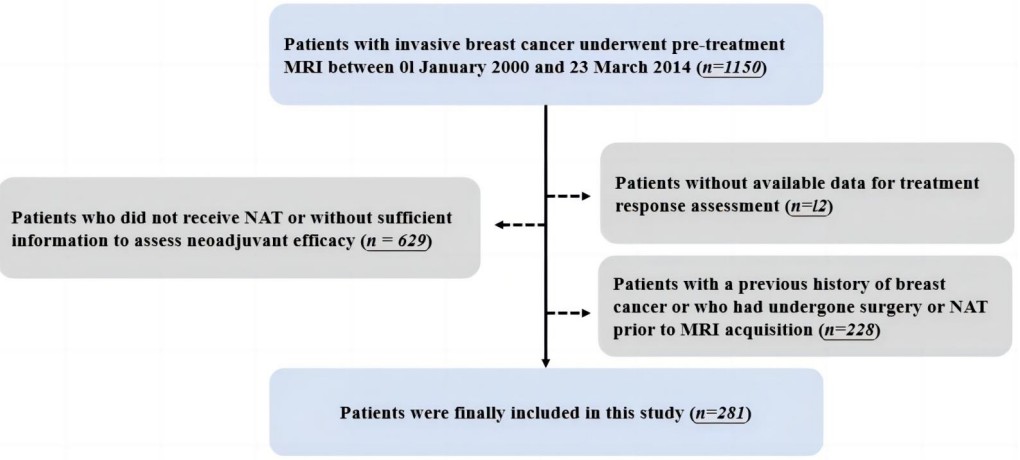

**Figure 1   Workflow for the inclusion and exclusion of patients.**

to build upon prior research that highlighted the predictive potential of pre-NAT DCE-MRI-based radiomic signatures for pCR in breast cancer by evaluating six different ML classifiers.

## METHODS

### Study population and data collection

We collected data for 1,150 consecutive patients with biopsy-proven invasive breast cancer between 01 January 2000 and 23 March 2014, all with pre-treatment MRI data available, from a publicly available dataset (Duke-Breast-Cancer-MRI) released in 2022 (*Saha et al., 2018*) from The Cancer Imaging Archive (TCIA, https://www.cancerimagingarchive.net/). Patients with a previous history of breast cancer or who had undergone surgery or NAT before MRI acquisition were excluded, resulting in a study population of 922 patients.

Clinical features (*e.g.*, patient age, menstrual status), MRI scanning parameters, tumor characteristics, response to NAT, and imaging characteristics (529 radiomic features extracted by in-house computer software from each of the 922 patients) were obtained for the 922 patients. Data obtained from TCIA did not require ethical approval and informed consent was waived since the TCIA dataset is comprised of de-identified patient information.

After excluding patients who did not receive NAT or without sufficient information to assess neoadjuvant efficacy ($n = 629$) and patients without available treatment response data ($n = 12$), 281 patients were finally included in this study. Detailed inclusion and exclusion procedures are shown in Fig. 1.

In this study, pCR was defined as the absence of residual invasive carcinoma (residual ductal carcinoma *in situ* is acceptable; ypT0/isN0), in the complete excision of breast specimen and ipsilateral sentinel lymph node or axillary lymph node dissection, after completion of NAT. The patients were categorized into two groups: those who achieved
(pCR) and those who did not (non-pCR). Subgroup analysis was mainly performed for the luminal and triple-negative (TN) subtypes of breast cancer. The luminal group includes luminal A (ER/PR positive, HER2 negative, and Ki-67 <14%) and luminal B (ER/PR positive, HER2 negative, and Ki-67 ≥ 14%; ER/PR positive, HER2 overexpression/amplification, regardless of Ki-67 expression) breast cancers, whereas TN breast cancer is defined as having a negative status for ER, PR, and HER2.

## MRI acquisition information

Breast MRI images were acquired using a 1.5T or 3.0T scanner in prone positions and consisted of a non-fat saturated T1-weighted sequence, a fat-saturated gradient-echo T1-weighted precontrast, and 3–4 postcontrast series in the axial plane. Pre-treatment DCE-MRI sequences were used for image analysis.

## Image postprocessing

Image annotation, segmentation, and feature extraction were completed as detailed in a previously published article (Saha et al., 2018).

### Image annotation and segmentation

Images were annotated using a three-dimensional (3D) box by one of eight fellowship-trained breast radiologists. One radiologist analyzed each case according to the pre-contrast, first post-contrast images, and subtraction images of both. Subsequently, four masks of the tumor, breast, and fibroglandular tissue (FGT) from pre-contrast and first post-contrast T1 weighted image (T1WI) sequences were extracted for each patient. Tumor masks were obtained from fuzzy C-means automated segmentation based on 3D annotations. Breast and FGT masks were automatically extracted from N4-corrected pre-contrast and first post-contrast T1WI sequences.

### Radiomic feature extraction

For each patient, a total of 529 radiomic features were extracted. These features were obtained from the literature and developed by the Duke-Breast-Cancer-MRI laboratory, all of which have been shown to have effective predictive power. These features were divided into 10 groups that covered various aspects of quantitative imaging, such as breast, tumor (including size, shape, texture, and enhancement), and FGT. Further details are provided in Fig. S1.

## Feature selection and model building

The Synthetic Minority Oversampling Technique (SMOTE) was used to oversample the minority data in the dataset of our study to eliminate the class imbalance of the original dataset and ensure model stability. A stratified random sampling method was then used to randomly assign the resulting dataset to training and validation groups at a ratio of 7:3. The SMOTE and stratified random sampling methods were also applied in the subgroup analyses of luminal breast cancer and TN breast cancer, respectively. All extracted radiomic features were normalized using $Z$-scores and converted to new scores [mean 0, standard deviation (SD) 1]. To eliminate redundant features and reduce overfitting or bias, it is necessary to select the optimal predictive features and improve the accuracy of the established model.
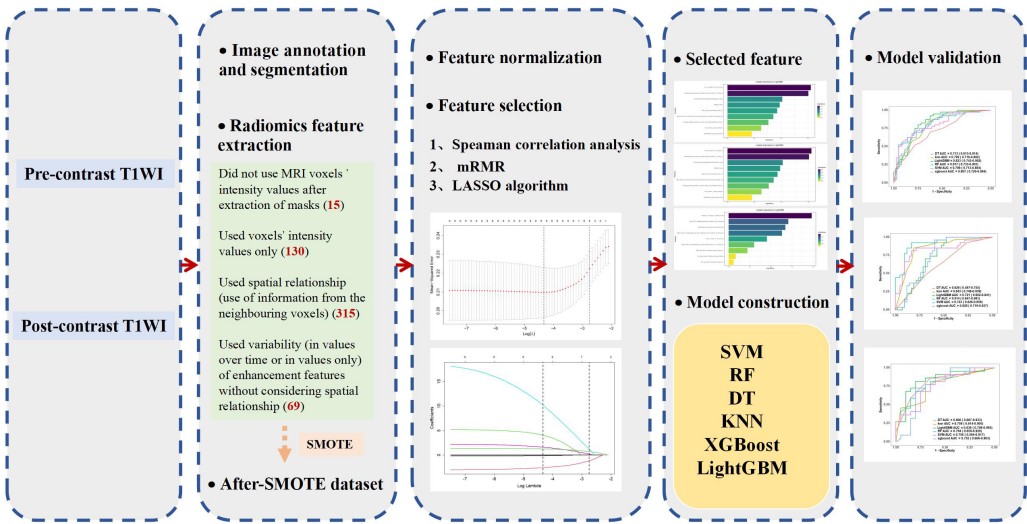

**Figure 2** Flowchart of the model construction.

After normalization, Spearman correlation analysis was used to reduce dimensionality and select features, and features with coefficients greater than 0.8 or $p$-values less than 0.05 in the training group were removed. The minimal-redundancy-maximal-relevance (mRMR) algorithm was then used to select the most relevant features for pCR prediction and to reduce redundancy between these features. As a result, the 10 most relevant and least redundant features were selected. The L1 regularization–based Least Absolute Shrinkage and the Selection Operator (LASSO) algorithm was used to select optimal predictive features and radiomic features with non-zero coefficients were selected for subsequent analysis of ML classification.

## Machine learning classifiers

We used six ML classifiers to establish radiomic models in the training group, which were then confirmed in the validation group. These six classifiers were as follows: support vector machine (SVM), random forest (RF), decision tree (DT), k-nearest neighbor (KNN), extreme gradient boosting (XGBoost), and light gradient boosting machine (LightGBM).

All feature classifier algorithms were implemented using the ML Python package (*i.e.,* Scikit-learn). The six classifiers were trained by grid search using ten-fold cross-validation in the training group to optimize the hyperparameters, and their performance was then tested in the validation group. The model-building process is presented in Fig. 2.

## Statistical analysis

All statistical analyses were completed using SPSS (version. 23.0) and Python (version. 2.7.1) software. A $p$-value less than 0.05 was considered statistically significant.

The area under the receiver operating characteristic (ROC) curve (AUC) was used to evaluate the performance of each classifier. The accuracy, sensitivity, and specificity of the optimal cutoff value were also reported. In addition, 95% confidence intervals

**Table 1  Patient information.**

| Characteristics | Total patients ($n = 281$) | Non-pCR group ($n = 217$) | pCR group ($n = 64$) | p-value |
|---|---|---|---|---|
| **Age** | 48.842 ± 10.880 | 49.395 ± 10.733 | 46.968 ± 11.250 | 0.117 |
| **Menopause (at diagnosis)** | | | | 0.345 |
| Premenopausal | 161 (57.30%) | 119 (54.84%) | 42 (65.62%) | |
| Postmenopausal | 119 (42.35%) | 97 (44.70%) | 22 (34.38%) | |
| NA | 1 (0.36%) | 1 (0.46%) | 0 (0.00%) | |
| **ER** | | | | **0.002** |
| positive | 164 (58.36%) | 145 (60.42%) | 38 (59.38%) | |
| negative | 117 (41.64%) | 95 (39.58%) | 26 (40.62%) | |
| **PR** | | | | **<0.001** |
| positive | 130 (46.26%) | 116 (48.33%) | 50 (78.12%) | |
| negative | 151 (53.74%) | 124 (51.67%) | 14 (21.88%) | |
| **HER2** | | | | **0.009** |
| positive | 80 (28.47%) | 64 (26.67%) | 37 (57.81%) | |
| negative | 201 (71.53%) | 176 (73.33%) | 27 (42.19%) | |
| **Molecular subtype** | | | | **0.004** |
| Luminal | 171 (60.85%) | 152 (63.33%) | 29 (45.31%) | |
| HER2-positive | 28 (9.96%) | 23 (9.58%) | 12 (18.75%) | |
| Triple-negative | 82 (29.18%) | 65 (27.09%) | 23 (35.94%) | |

Notes.

Data are represented as mean ± standard deviation (SD) or number of patients, with percentages in parentheses. Bold $p$ values indicate statistical significance ($p < 0.05$).

ER, estrogen receptor; PR, progesterone receptor; HER2, human epidermal growth factor receptor 2; NA, not available.

(CI) were estimated using bootstrapping with 1,000 samples. Differences between groups for continuous variables were assessed using the Mann–Whitney $U$ test or Student's $t$-test. Differences between groups were assessed using the Chi-square test for categorical variables.

# RESULTS

## Patient information and pathological features

A total of 281 patients with a mean age of 48.842 ± 10.880 years (range 24–77 years) were analyzed and classified into pCR ($n = 64$) and non-pCR ($n = 217$) groups. The prevalence of molecular subtypes (ER, PR, and HER2 status) was significantly different between the pCR and non-pCR groups (all $p < 0.009$). Age ($p = 0.117$) and menstrual status ($p = 0.345$) were not significantly different between the pCR and non-PCR groups. Patient characteristics are presented in Table 1.

## Performance of the radiomic models with machine learning classifiers

Nine radiomic features [F1_DT_T1NFS (T11 = 0.2,T12 = 0.8), Grouping_based_variance _of_washin_slope_3D_tissue_T1_Group_1, Enhancement Cluster Neighborhood Similarity_Tumor, BEDR2_Tumor, SER_map_Autocorrelation_tissue_T1, Ratio_Tissue_ vol_enhancing_more_than_20percent_from_PostCon_to_Breast_Vol, Grouping_based_

proportion_of_tumor_voxels_2D_tumorSlice_Group_1, Sum_variance__tissue_PostCon, WashinRate_map_kurtosis_tumor] were used to construct the pCR prediction models. The models were constructed with 6 ML classifiers after mRMR and L1 regularization–based LASSO selection (Fig. 3A).

Table 2 summarizes the diagnostic performance of the radiomics models for predicting pCR after NAT in the training and validation groups.

In the training group, the AUCs of SVM, RF, DT, KNN, XGBoost, and LightGBM were 0.778, 1, 0.867, 1, 0.964, and 0.773, respectively. In the validation group, AUC values ranged from 0.767 to 0.823. The best performance was obtained using LightGBM (AUC 0.823, 95% CI [0.743–0.902], accuracy 74.0%, sensitivity 85.0%, specificity 67.2%), followed by RF (AUC 0.816, 95% CI [0.731–0.902], accuracy 75.0%, sensitivity 65.0%, specificity 81.3%).

The ROC curves and variable importance of the LightGBM model are displayed in Fig. 4A.

## Subgroup analysis of radiomics model performance for predicting pCR in TN and luminal breast cancer

Table 3 presents the diagnostic performance of the radiomics models in predicting pCR after NAT in two subgroups.

Out of 171 cases of luminal breast cancer, 29 patients achieved pCR while 142 did not. Analysis in the validation group revealed that the RF algorithm had the best performance in predicting pCR after NAT with an AUC of 0.914 (95% CI [0.847–0.981]) (accuracy 87.0%, sensitivity 85.2%, and specificity 88.1%; Fig. 4B). The final selected radiomic features for luminal breast cancers are shown in Fig. 3B.

Among the 82 patients with TN breast cancer, 23 patients achieved pCR and 59 did not. In the validation group, the LightGBM algorithm performed best in predicting pCR after NAT with an AUC of 0.836 (95% CI [0.708–0.965]) (accuracy 78.6%, sensitivity 68.2%, and specificity 90.0%; Fig. 4C). The final selected radiomic features for TN breast cancers are shown in Fig. 3C.

## DISCUSSION

In this study, we investigated the efficacy of noninvasive radiomics models that utilize the N4-corrected pre-contrast and first post-contrast T1WI images taken before treatment for predicting pCR in breast cancer. Our analysis demonstrates that prediction models developed using LightGBM and RF algorithms yield excellent predictive performance, with AUCs of 0.823 and 0.816 in the validation group, respectively. During subgroup analysis, RF had higher AUC and accuracy than the other five classifiers in predicting pCR for luminal breast cancer, while LightGBM yielded higher AUC and accuracy than the other five classifiers in predicting pCR for TN breast cancer.

Recent studies have investigated the value of DCE-based radiomics models in predicting pCR in patients before, during, and after NAT (Liu et al., 2019; O'Donnell et al., 2022). NAT can improve breast-conservation rate, and meanwhile, patients who achieve pCR have higher OS and DFS than those without (Cho et al., 2014). While NAT effectiveness can

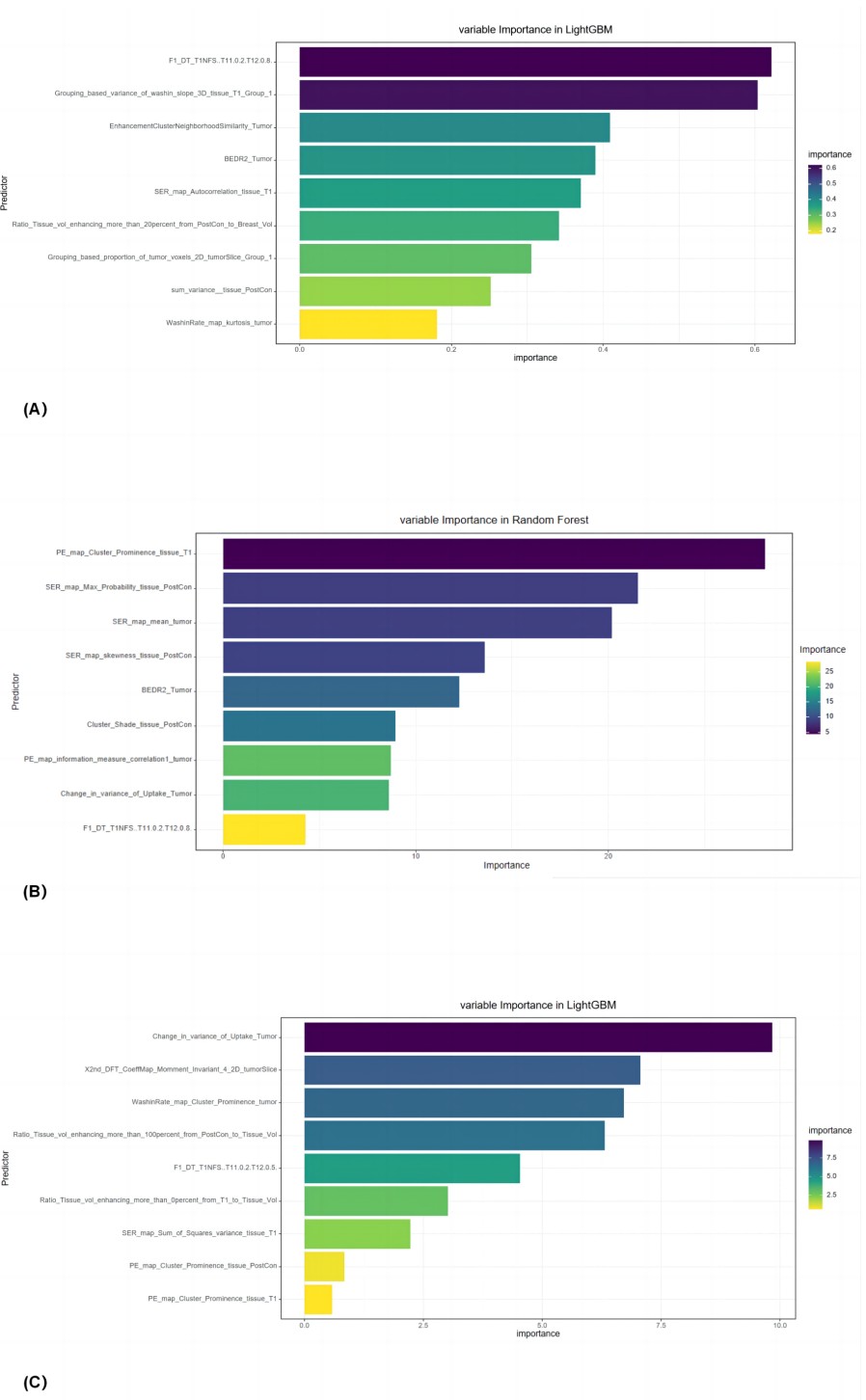

**Figure 3** The final selected radiomic features for (A) all breast cancers, (B) luminal, and (C) triple-negative breast cancers.

**Table 2  Diagnostic performance of radiomics models for six classifiers in training and test groups.**

| Models | Training group | | | | Test group | | | |
|---|---|---|---|---|---|---|---|---|
| | AUC | Accuracy | Sensitivity | Specificity | AUC | Accuracy | Sensitivity | Specificity |
| SVM | 0.778 | 0.290 | 0.102 | 0.399 | 0.787 | 0.317 | 0.175 | 0.406 |
| KNN | 1 | 1 | 1 | 1 | 0.782 | 0.673 | 0.625 | 0.703 |
| DT | 0.867 | 0.809 | 0.728 | 0.859 | 0.767 | 0.730 | 0.630 | 0.810 |
| RF | 1 | 1 | 1 | 1 | 0.816 | 0.750 | 0.650 | 0.813 |
| XGBoost | 0.964 | 0.900 | 0.886 | 0.909 | 0.785 | 0.712 | 0.575 | 0.797 |
| LightGBM | 0.773 | 0.701 | 0.852 | 0.614 | 0.823 | 0.740 | 0.850 | 0.672 |

Notes.
AUC, the area under the receiver operating characteristic curve; SVM, support vector machine; RF, random forest; DT, decision tree; KNN, k-nearest neighbor.

be evaluated before surgery using RECIST version 1.1, confirmation of efficacy can only be done postoperatively using the Miller-Payne grading system. Additionally, patients with poor response to NAT are at risk of severe drug toxicity and delayed surgical treatment. As such, the application of radiomics is crucial for accurate non-invasive prediction of the response to NAT before therapy selection.

Several studies have employed ML classifiers to build radiomics models for predicting pCR based on pre-treatment MRI (*Liu et al., 2019*; *Li et al., 2021*; *Vicent et al., 2022*; *Chen et al., 2020*). However, findings are contradictory due to the use of different radiomic features, ML classifiers, and prediction models. *Li et al. (2021)* used 23 ML classifiers to construct radiomics models based on multi-phase DCE-MRI images for early prediction of pCR after NAT in HER2-positive breast cancer patients and compared the performance of the classifiers. They found that linear SVM based on multi-phase DCE (AUC = 0.84) was superior to logistic regression models using first post-contrast T1WI images (AUC = 0.69) (*Li et al., 2021*). However, the study lacked an independent validation group. *Vicent et al. (2022)* used 10 ML classifiers to construct pCR prediction models for NAT based on perfusion/diffusion imaging biomarkers and radiomic features extracted from pre-treatment multiparametric MRI. In contrast, they found that quadratic discriminant analysis (QDA) based on imaging features yielded the highest accuracy (87.5%) (*Vicent et al., 2022*). Previous studies on radiomics for predicting pCR in breast cancer had smaller sample sizes ranging from 58 to 158 patients (*Vicent et al., 2022*; *Chen et al., 2020*). Our present study enrolled 281 patients, providing a larger sample size. This study aimed to evaluate the performance of radiomics models based on pre-contrast and first post-contrast T1WI images in predicting pCR before NAT. We compared the prediction performance of six ML classifiers for pCR and selected the optimal classifier. In addition, the MRI parameters used varied between publicly available datasets, so we used only one dataset. A future prospective multicenter study with a larger sample should be considered.

This study extracted 529 radiomic features, which allowed for a comprehensive quantification of breast, FGT, and tumor characteristics. The predictive performance of these features was shown to be good (*Cain et al., 2019*; *Saha et al., 2018*). We included nine radiomic features to build the model to predict pCR after NAT in breast cancer. The nine features belong to the feature groups of tumor size and morphology, tumor

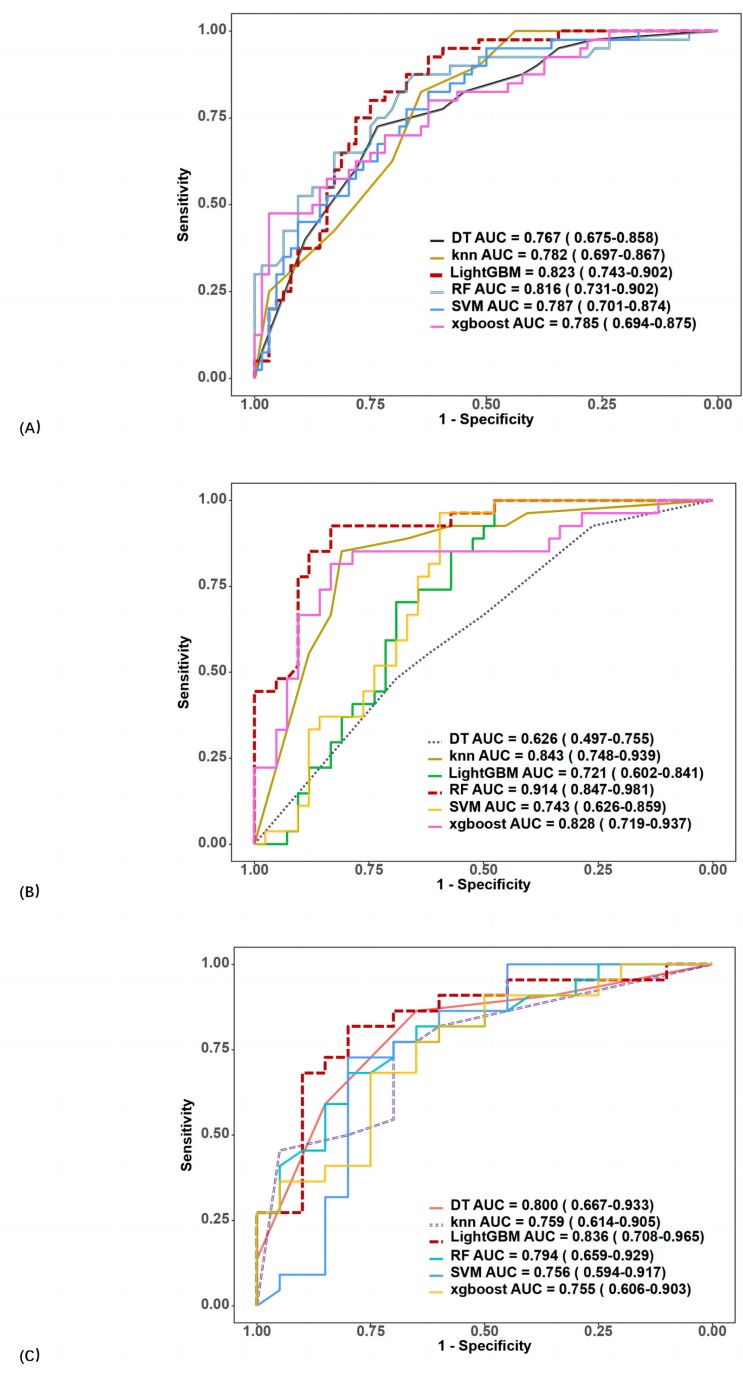

**Figure 4** ROC curves of six classifiers in the radiomics model for predicting pCR (A), including (B) luminal, and (C) triple-negative breast cancer in the validation group.

enhancement variation, combining tumor and FGT enhancement, tumor enhancement spatial heterogeneity, FGT enhancement texture, and variation. In building the model this

**Table 3** Diagnostic performance of radiomics models for luminal and triple-negative breast cancers.

| Models | Training group | | | | Test group | | | |
|---|---|---|---|---|---|---|---|---|
| | AUC | Accuracy | Sensitivity | Specificity | AUC | Accuracy | Sensitivity | Specificity |
| Luminal type breast cancer | | | | | | | | |
| SVM | 0.897 | 0.838 | 0.950 | 0.770 | 0.743 | 0.681 | 0.778 | 0.619 |
| KNN | 1 | 1 | 1 | 1 | 0.843 | 0.754 | 0.889 | 0.667 |
| DT | 0.919 | 0.888 | 0.850 | 0.910 | 0.626 | 0.594 | 0.484 | 0.684 |
| RF | 1 | 1 | 1 | 1 | 0.914 | 0.870 | 0.852 | 0.881 |
| XGBoost | 0.998 | 0.988 | 0.967 | 1 | 0.828 | 0.797 | 0.741 | 0.833 |
| LightGBM | 0.899 | 0.869 | 0.883 | 0.860 | 0.721 | 0.623 | 0.556 | 0.667 |
| Triple negative breast cancer | | | | | | | | |
| SVM | 0.917 | 0.079 | 0.125 | 0.051 | 0.756 | 0.381 | 0.546 | 0.200 |
| KNN | 1 | 1 | 1 | 1 | 0.759 | 0.738 | 0.773 | 0.700 |
| DT | 0.932 | 0.889 | 0.815 | 0.914 | 0.800 | 0.714 | 0.813 | 0.654 |
| RF | 1 | 1 | 1 | 1 | 0.794 | 0.738 | 0.682 | 0.800 |
| XGBoost | 0.989 | 0.968 | 0.958 | 0.974 | 0.755 | 0.691 | 0.636 | 0.750 |
| LightGBM | 0.910 | 0.825 | 0.875 | 0.795 | 0.836 | 0.786 | 0.682 | 0.900 |

Notes.
AUC, the area under the receiver operating characteristic curve; SVM, support vector machine; RF, random forest; DT, decision tree; KNN, k-nearest neighbor.

way, we confirmed the significance of pre-NAT tumor and FGT MRI features to assess pCR.

LightGBM yielded the highest performance (AUC = 0.823) in predicting pCR after NAT in breast cancer compared to the other five classifiers (AUC = 0.767–0.816). LightGBM is a gradient-boosting algorithm developed to address the drawbacks of traditional gradient-boosting methods, such as low efficiency and poor scalability (*Ke et al., 2017*). It is known to have high accuracy, fast training, and good model stability, making it suitable for processing large-scale data. It is considered an improved gradient-boosting algorithm and has demonstrated excellent performance in various applications. A previous study investigated the predictive efficacy of radiomics based on pre- and post-contrast T1WI for pCR (*Cain et al., 2019*). However, only two ML algorithms (logistic regression and support vector machine) were used, yielding AUCs of 0.658 and 0.593, respectively. This suggests that LightGBM exhibits superior predictive ability and should be used in constructing effective radiomics prediction models.

Subgroup analysis of luminal breast cancer showed that RF had the highest predictive performance and accuracy, indicating its usefulness in predicting pCR for this type of cancer. RF is a DT-based ensemble learning method capable of processing high-dimensional data with higher accuracy than a single classifier and has proven valuable in predicting breast cancer prognostic biomarkers and molecular subtypes. LightGBM had better predictive ability and accuracy in the TN breast cancer subgroup analysis compared to the other five classifiers. To our knowledge, few studies have compared the value of different ML classifiers in predicting pCR in breast cancer, or in various subtypes of breast cancer. Our results indicated that the choice of ML classifiers is essential for constructing pCR

prediction models and that radiomics models based on pre- and post-contrast T1WI images before treatment have potential value in predicting pCR in breast cancer treatment.

The pCR rate observed in this study was 22.8% (64/281), which is consistent with previous literature (*Precht et al., 2010*; *Sachelarie et al., 2006*). The definition of pCR can influence its reported rate, and the inclusion or exclusion of residual DCIS remains a topic of debate. *Osdoit et al. (2022)* demonstrated no significant differences in 3-year event-free survival, distant recurrence-free survival, or locoregional recurrence between pCR patients with residual DCIS and those without. This study defined pCR as the absence of invasive carcinoma following NAT, regardless of DCIS presence. This definition aligns with that used in most studies (*Spring, Bar & Isakoff, 2022*).

### Limitations

This is a retrospective study conducted at a single center, which may have introduced selection bias and limited the generalizability of the results to other populations or settings. Future studies using multi-center data and external validation sets are needed to validate the generalization capabilities of the model. Further, only pre- and post-contrast T1-weighted images were used. Due to the small number of HER2-positive breast cancers, only TN and luminal breast cancers were included in the subgroup analysis. Additionally, the MR images of 281 patients with breast cancer came from different MR scanners with distinct imaging parameters. Finally, our study included different NAT treatment regimens, including neoadjuvant chemotherapy, endocrine and anti-HER2/neu antibody therapy, and subgroup analysis based on treatment regimens was not performed.

## CONCLUSION

Our study indicates that the LightGBM model along with pre- and post-contrast T1WI radiomic features can aid in predicting pCR after NAT for breast cancer and may have clinical applications. RF and LightGBM are recommended for predicting pCR in luminal and TN breast cancers, respectively. Further studies are warranted to increase the robustness of our findings.

### Funding
The authors received no funding for this work.

### Competing Interests
The authors declare there are no competing interests.

### Author Contributions
- Xue Li conceived and designed the experiments, performed the experiments, analyzed the data, prepared figures and/or tables, and approved the final draft.
- Chunmei Li performed the experiments, analyzed the data, prepared figures and/or tables, and approved the final draft.

- Hong Wang performed the experiments, prepared figures and/or tables, and approved the final draft.
- Lei Jiang performed the experiments, analyzed the data, prepared figures and/or tables, authored or reviewed drafts of the article, and approved the final draft.
- Min Chen conceived and designed the experiments, analyzed the data, prepared figures and/or tables, authored or reviewed drafts of the article, and approved the final draft.

## Data Availability

The Duke-Breast-Cancer-MRI dataset is available at The Cancer Imaging Archive (TCIA): https://www.cancerimagingarchive.net/collection/duke-breast-cancer-mri/.

## Supplemental Information

Supplemental information for this article can be found online at http://dx.doi.org/10.7717/peerj.17683#supplemental-information.

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
