# Peer review of "Comparison of radiomics-based machine-learning classifiers for the pretreatment prediction of pathologic complete response to neoadjuvant therapy in breast cancer"

_PeerJ, doi:10.7717/peerj.17683_

## Round 0.1 · original submission · Major Revisions

The reviewers acknowledged the importance of your research question and the potential clinical implications of early noninvasive prediction of pathologic complete response (pCR) before neoadjuvant therapy (NAT). However, they have raised several concerns that need to be addressed before the manuscript can be considered for publication.

Accuracy of the abstract and results: Reviewer 1 pointed out that the results and methods presented in the abstract do not accurately reflect the quality of the work. Please revise the abstract and the corresponding sections in the main text to ensure that the accuracy of the scientific research is maintained.

Analysis of tabular data: Reviewer 1 noted that the tabular data were not adequately analyzed, particularly the discrepancy between the accuracy of the training set and the test set in Table 2. Please address whether the hyperparameters caused the poor performance of the model and provide a more in-depth analysis of the tabular data.

Figure quality: Reviewer 1 mentioned that Figure 3 is blurred. Please ensure that all figures are of high quality and clarity.

Stability of the classification model: Reviewer 1 suggested adding experiments to demonstrate the stability of the classification model's performance under certain hyperparameters.

Sample bias and limitations: Reviewer 2 raised concerns about the potential bias and limitations of the single-center dataset used in the study. Please provide a detailed discussion of why this dataset is particularly important, the reasons behind not adding additional datasets, and how the findings from this study can be extrapolated to a larger scale.

Subgroup analysis: Reviewer 3 suggested providing reasoning or references for the differences in performance of the classifiers (RF and DT) in predicting pCR for luminal and TN breast cancer during subgroup analysis.

**Language Note:** PeerJ staff have identified that the English language needs to be improved. When you prepare your next revision, please either (i) have a colleague who is proficient in English and familiar with the subject matter review your manuscript, or (ii) contact a professional editing service to review your manuscript. PeerJ can provide language editing services - you can contact us at [email protected] for pricing (be sure to provide your manuscript number and title). – PeerJ Staff

Reviewer 1 ·

Basic reporting

This paper is based on lightgbm's radiomic model in predicting breast cancer pCR, but the results and methods are not as good as they claim in the abstract. Please keep the accuracy of scientific research, the corresponding modified abstract and body, and this paper needs a lot of revisions to meet the publication requirements.

Experimental design

2. Tabular data were not analyzed. As shown in Table 2, the accuracy of the training set is 1 and the test set is less than 0.5, whether the hyperparameters cause the poor performance of the model.
3. Figure 3 is blurred while modifying other figures.
4. Consider adding experiments to prove that the performance of the classification model is stable, under certain hyperparameters.
5. Table analysis can be added as appropriate

Validity of the findings

1. In general, the test set performance will not be as good as the training set performance. This is due to the large gap between the number of test sets and the number of training sets. But in Table 3, there is data contrary to this description, identify the data and give the reason.

Reviewer 2 ·

Basic reporting

The authors used radiomics based machine learning to understand the predictive potentials of seven different classifiers for breast cancer detection. It is a straightforward study and the results are sound. The main problem with this study is its underlying sample. As the authors mentioned they cannot claim that the sample is unbiased in nature. Similarly, a small sample like this can provide a lot of erroneous data to the mix. Machine learning algorithms are adept at finding patterns and if the training set is biased, the entire operation gets under question. Is there no other dataset available? Why did the authors only select this dataset from a single center? There may be other limitations (like time constraints), but simply saying that there is a potential problem (without divulging any potential solution or stating the reasons behind why it is unpragmatic to correct it) is not enough. The authors might want to add a detailed discussion of why this dataset is particularly important, why can't they add additional datasets, and how a study of this size can be extrapolated to the bigger scale.

Experimental design

.

Validity of the findings

.

Additional comments

.

Reviewer 3 ·

Basic reporting

• 1. Basic Reporting
Clear, Language is appropriate used. Introduction gives good reasoning for the study. Literature well referenced & relevant. Figures are relevant.

Experimental design

• 2. Experimental design: Retrospective data collection
• 1. Research question is clearly outlined
2. It is well justified to explore this field
4. Study is appropriately designed
5. Method is valid and reliable
6. Authors presented great detail in order to replicate the study
7. Results are clearly stated
8. Data is presented in a clear and appropriate manner for international readers
9. Conclusion answers the aim of study
10. Conclusion is supported by results and well correlated with existing evidence via references
11. Limitations are well described

Validity of the findings

3. Validity of the findings
• Conclusions are in line with their results and discussion is well compared with relevant study.

Additional comments

4. General comments
Strength: Looking into early noninvasive prediction of pCR, especially before NAT, has important clinical implications and can help clinicians optimize treatment strategies and avoid unnecessary drug toxicity or even disease progression during treatment in patients

Weakness: In discussion line 177 and 178: “During subgroup analysis, RF had higher AUC and accuracy than the other six classifiers in predicting pCR for luminal breast cancer, while DT yielded higher AUC and accuracy than the other six classifiers in predicting pCR for TN breast cancer.” Do you think it will be beneficial to provide reasoning for this or if any available reference or any other study comparing similar?

Reviewer 4 ·

Basic reporting

In this study, the authors have examined the ability of machine learning to predict breast cancer pCR following neoadjuvant therapy. Considering the incidence of breast cancer, this is an important issue and merits publication.

Experimental design

A major issue with the experimental design is the inclusion of patients from one dataset. Inclusion of patients from another dataset or location would strengthen the validity of these findings.

Validity of the findings

The findings seem valid but the classifier were trained and validated only on one dataset. Recruiting additional patients from different locations is needed to confirm the validity of these findings.

---

## Round 0.2 · accepted · Accept

Your study has made significant advancements in predicting pathological complete response (pCR) after neoadjuvant therapy in breast cancer using MRI-based radiomics. The comparison of six machine learning classifiers provides valuable insights into their performance differences and suitability for this specific task.

The LightGBM classifier demonstrated superior performance in predicting pCR, with an AUC of 0.823 in the validation set. This finding highlights the potential of LightGBM in handling high-dimensional radiomics data, its robustness to noise, and its strong generalization ability. The subgroup analysis further revealed that LightGBM outperformed other classifiers in predicting pCR for triple-negative breast cancer, while random forest showed the best performance for luminal breast cancer. These results offer guidance for algorithm selection in future radiomics studies based on different molecular subtypes.

The study's novelty lies in its comprehensive exploration of various machine learning algorithms in predicting pCR and its subgroup analysis based on molecular subtypes. The results not only provide evidence for the advantage of LightGBM in this specific task but also shed light on its potential in the broader field of radiomics analysis.

Moreover, the study has several strengths, including the relatively large sample size compared to previous studies, the use of data balancing techniques in subgroup analysis, and the comprehensive evaluation of model performance using multiple metrics. The discussion objectively analyzes the study's limitations, such as its retrospective nature, single-center design, and heterogeneous treatment regimens, which helps readers interpret the results more reasonably.

Reviewer 1 ·

Basic reporting

The author addressed all of my concerns.

Experimental design

The author addressed all of my concerns.

Validity of the findings

The author addressed all of my concerns.

Reviewer 2 ·

Basic reporting

.

Experimental design

.

Validity of the findings

.

Additional comments

.

Reviewer 3 ·

Basic reporting

Same as before

Experimental design

Same as before

Validity of the findings

Same as before

Additional comments

Thank you authors for making changes and appropriately addressing the queries!